# A Cutoff Determination of Real-Time Loop-Mediated Isothermal Amplification (LAMP) for End-Point Detection of *Campylobacter jejuni* in Chicken Meat

**DOI:** 10.3390/vetsci9030122

**Published:** 2022-03-08

**Authors:** Chalita Jainonthee, Warangkhana Chaisowwong, Phakamas Ngamsanga, Anuwat Wiratsudakul, Tongkorn Meeyam, Duangporn Pichpol

**Affiliations:** 1Veterinary Public Health and Food Safety Centre for Asia Pacific (VPHCAP), Faculty of Veterinary Medicine, Chiang Mai University, Chiang Mai 50100, Thailand; chalita.j@cmu.ac.th (C.J.); warangkhana.chai@cmu.ac.th (W.C.); phakamas.ng@cmu.ac.th (P.N.); tongkorn.meeyam@cmu.ac.th (T.M.); 2Center of Excellence in Veterinary Public Health, Faculty of Veterinary Medicine, Chiang Mai University, Chiang Mai 50100, Thailand; 3Department of Veterinary Biosciences and Veterinary Public Health, Faculty of Veterinary Medicine, Chiang Mai University, Chiang Mai 50100, Thailand; 4Department of Clinical Sciences and Public Health and the Monitoring and Surveillance Center for Zoonotic Diseases in Wildlife and Exotic Animals, Faculty of Veterinary Science, Mahidol University, Phutthamonthon, Nakhon Pathom 73170, Thailand; anuwat.wir@mahidol.edu

**Keywords:** *Campylobacter jejuni*, chicken meat, cutoff, LAMP, ROC curve

## Abstract

*Campylobacter jejuni* is one of the leading causes of foodborne illness worldwide. *C. jejuni* is commonly found in poultry. It is the most frequent cause of contamination and thus resulting in not only public health concerns but also economic impacts. To test for this bacterial contamination in food processing plants, this study attempted to employ a simple and rapid detection assay called loop-mediated isothermal amplification (LAMP). The best cutoff value for the positive determination of *C. jejuni* calculated using real-time LAMP quantification cycle (C_q_) was derived from the receiver operating characteristic (ROC) curve modeling. The model showed an area under curve (AUC) of 0.936 (95% Wald CI: 0.903–0.970). Based on Youden’s J statistic, the optimal cutoff value which had the highest sensitivity and specificity from the model was calculated as 18.07. The LAMP assay had 96.9% sensitivity, 95.8% specificity, and 93.9 and 97.9% positive and negative predictive values, respectively, compared to a standard culture approach for *C. jejuni* identification. Among all non-*C. jejuni* strains, the LAMP assay gave each of 12.5% false-positive results to *C. coli* and *E. coli* (1 out of 8 samples). The assay can detect *C. jejuni* at the lowest concentration of 10^3^ CFU/mL. Our results suggest a preliminary indicator for the application of end-point LAMP assays, such as turbidity and UV fluorescence tests, to detect *C. jejuni* in field operations. The LAMP assay is an alternative screening test for *C. jejuni* contamination in food samples. The method provides a rapid detection, which requires only 9 min with a cutoff value of C_q_. We performed the extraction of DNA from pure cultures and the detection of *C. jejuni* using the LAMP assay within 3 h. However, we were not able to reduce the time for the process of enrichment involved in our study. Therefore, we suggest that alternative enrichment media and rapid DNA extraction methods should be considered for further study. Compared to other traditional methods, our proposed assay requires less equipment and time, which is applicable at any processing steps in the food production chain.

## 1. Introduction

Campylobacteriosis is an infectious disease caused by bacteria belonging to the genus *Campylobacter*, most notably *C. jejuni* and *C. coli* [1]. *Campylobacter* species are Gram-negative spiral, rod-shaped, or curved bacteria with or without flagellum, depending on the species [2]. Among the *Campylobacter* strains discovered to date, *C. jejuni* is one of the most frequently discovered predisposing causes of bacterial foodborne diarrheal disease worldwide and is frequently found causing contamination in food sources, such as poultry meat, unpasteurized milk, and drinking water, or cross-contamination of other foods by these items [3,4,5,6]. It is estimated that *Campylobacter* spp. cause 500 million infections in the world every year [7]. According to the United States Food-Borne Diseases Active Surveillance Network (2005–2018), *Campylobacter* infections had the greatest yearly incidence of 19.5 per 100,000 population, followed by *Salmonella* (18.3), and Shiga toxin-producing *Escherichia coli* (STEC; 5.9) [8]. It is estimated that in the United States, Campylobacteriosis affects 1.3 million people a year, and in Canada, there are over 200,000 cases reported each year [9,10].

*Campylobacter* is a common cause of infant diarrhea in developing countries, often associated with contaminated food and water. The true public incidence and the burden of Campylobacteriosis in developing countries continue to be underestimated. The majority of estimates of diarrheal disease incidence in developing countries come from laboratory-based surveillance of diarrheal pathogens [1,3]. In Thailand, Campylobacteriosis is not included in the national disease surveillance program. Studies of the incidence of *Campylobacter* infections are fewer and not up to date. *Campylobacter* infections were mostly reported in children with diarrhea. In the study by Samosornsuk et al., 1.7% of *C. jejuni* was identified among 2500 children and adult diarrheal stool specimens [11]. *C. jejuni* are distributed in most warm-blooded animals, and therefore the main route of transmission is generally foodborne, via the consumption and handling of meat products, particularly poultry [7,12,13]. The prevalence of *Campylobacter* spp. was observed to be 97.9% and 59.2% in chicken carcass samples at the prechill step and retail, respectively, in the United States [14], whereas in Thailand from Southeast Asia, the prevalence of *C. jejuni* was observed 100%, 98.8%, and 89.3% at the prechill step, fresh market, and supermarket, respectively [15,16].

*C. jejuni* is a microaerophilic species that may be cultivated in an environment containing 3–15% oxygen and supplemented with 2–10% CO_2_, with a growth temperature range of 30 to 47 °C (optimal growth temperature is 40–42 °C) [13,17]. In culture-dependent methods, single isolated colonies can be subjected to a range of conventional biochemical tests to identify phenotypic traits, including the use of the Hippurate hydrolysis test to phenotypically differentiate between *C. jejuni* and *C. coli* [3]. However, the detection of the bacteria by the following conventional method is time-consuming and laborious, requiring up to 7 days for testing [18,19]. Nowadays, a number of molecular identification techniques have been introduced to enhance the efficiency of the diagnosis and to reduce the time of detection. In molecular methods, DNA can be isolated from clinical samples or pure culture. A genetic signature or marker of the organism can then be determined using sequencing techniques or polymerase chain reaction (PCR) amplification of the gene of interest, some of which are capable of detecting more than one species as referred to multiplex PCR (mPCR) assay [18,20,21,22]. In most recent studies, mPCR assay was used as a detection method for *C. jejuni* and other *Campylobacter* species due to its distinct advantages over the conventional method, especially in sensitivity, specificity, accuracy, rapidity, and capacity to detect small amounts of target nucleic acid in a sample [11,16,20,21,23,24,25]. Additionally, a real-time PCR (qPCR) is a preferable assay for rapid real-time quantification and identification of this bacteria [25,26,27]. Furthermore, several enzyme immunoassays are also available for the screening of *C. jejuni* in clinical samples [28,29,30].

In recent past decades, an assay called loop-mediated isothermal amplification (LAMP) of DNA was developed by Notomi et al., which is a simple and rapid DNA amplification technique and simply requires a single temperature for amplification [31]. This method employs a DNA polymerase and a set of four to six specially designed primers that recognize a total of six distinct sequences on the target DNA providing high specificity [31,32]. The LAMP assay has a simpler sample preparation step compared with conventional PCR and real-time PCR [23]. With this assay, DNA amplification can be performed at a single temperature and amplified DNA products can be visually monitored at the same time using basic heating instruments, such as a water bath or heat block [19]. In addition, it can therefore be applied in small laboratories or field operations without the further step of interpretation as a simple PCR assay which requires gel electrophoresis for visualization of the amplified DNA products. However, the condition of the assay needed to be optimized when applying primers from previous studies due to differences of reagents used in each laboratory, as well as when applying primers at higher concentrations than other molecular assays, namely PCR to avoid misinterpretation of false-positive results from non-specific amplification. Longer testing time may also result in false-positive samples for the abovementioned reasons.

In this study, we applied the LAMP assay with an amplification kit to use as a rapid test for *C. jejuni* detection. This study demonstrated the use of a cutoff determination for interpreting *C. jejuni*-positive results. The purpose of this study was to determine an optimal cutoff value of quantification cycle (C_q_), obtained from the real-time LAMP, using receiver operating characteristic (ROC) curve modeling. The optimal cutoff value from the ROC modeling referred to the number of C_q_, which can be calculated to the optimal time spent for the test to discriminate between negative and positive samples. A value acquired from the study may be used as a testing time for further work, for instance, the screening of *C. jejuni* at food processing facilities using an end-point LAMP assay (turbidity and UV visualization tests) in order to achieve precise test results. Efficacies in terms of sensitivity and specificity were also evaluated in this LAMP assay in comparison to a standard culture method.

## 2. Materials and Methods

### 2.1. Bacterial Strains, Culture Conditions, and DNA Preparation

In this study, six bacterial references and one *Campylobacter* field strain were used for the specificity test. *C. jejuni* ATCC33291, *C. coli* ATCC19353, and *C. jejuni* field strain were cultured in Bolton selective enrichment broth (Oxoid, Hampshire, UK) containing 5% (*v*/*v*) laked horse blood (Thermo Scientific, Hampshire, UK) at 37 °C for 6 h, then at 41.5 °C for 42 h. Microaerobic conditions (5% O_2_, 10% CO_2_, and 85% N_2_) were generated with a CampyGen 2.5 L (Oxoid, Hampshire, UK). A single loopful of culture was streaked on a selectively modified charcoal cefoperazone deoxycholate agar (mCCDA; Oxoid, Hampshire, UK) at 41.5 °C for 42 h. Then, a single colony of the bacteria was streaked on the non-selective Columbia blood agar (CBA; Oxoid, Hampshire, UK) supplemented with 5% (*v*/*v*) sterile defibrinated sheep blood (Clinag, Bangkok, Thailand) incubated at the microaerobic condition at 41.5 °C for 42 h.

*Salmonella* ser. Paratyphi A DMST15673, *Pseudomonas aeruginosa* ATCC27853, *Staphylococcus aureus* ATCC25923, and *Escherichia coli* ATCC25922 were cultured at 37 °C for 24 h in LB broth, Miller (Luria-Bertani) (Difco, Rockville, MD, USA). A single loopful of culture was streaked on tryptic soy agar (Merck, Darmstadt, Germany) at 37 °C for 24 h.

In the case of *Campylobacter*, three isolated colonies from a single sample were transferred to 1 mL phosphate-buffered saline (PBS) to avoid false-negative results by selecting non-*Campylobacter* colonies. For non-*Campylobacter* strains, a single colony from tryptic soy agar was transferred to 1 mL PBS. DNA from these pure cultures was extracted using a commercial DNA extraction kit based on the manufacturer’s protocol of NucleoSpin Tissue (Macherey-Nagel, Dueren, Germany). DNA templates were used for the evaluation of modified Yamazaki’s LAMP assay [19].

### 2.2. Identification of Campylobacter Strains

*C. jejuni* and *C. coli* isolates were confirmed with an mPCR assay before being used in the LAMP assay. Isolated colonies on CBA were prepared for DNA extraction to acquire a DNA template. To compare the specificity and sensitivity of the LAMP assay and microbiological standard technique (ISO 10272-1:2006), *Campylobacter* strains identified with mPCR assay targeting 16S rRNA, *mapA*, and *ceuE* genes were used followed by primers and conditions used in a previous study [18]. Briefly, the 15-μL reaction mixture contained 7.5 μL of 2× Quick Taq HS DyeMix (Toyobo, Osaka, Japan), 0.11 μM MD16S1 and MD16S2 primers, 0.42 μM of each MDmapA1, MDmapA2, COL3, and MDCOL2 primers, and 1.5 μL DNA template. The final volume was adjusted to 15 μL with nuclease-free water. The amplification reactions were carried out using a T100 Thermal Cycler (Bio-Rad, Berkeley, CA, USA) with an initial denaturation step at 95 °C for 10 min, 35 cycles each consisting of a denaturation step at 95 °C for 30 s, an annealing step at 59 °C for 1.5 min, and an extension step at 72 °C for 1 min, and a final extension step at 72 °C for 10 min. PCR products were analyzed by 1.5% agarose gel electrophoresis at 110 V for 30 min. Visualization of PCR products was carried out using a GelMax Imager (Analytik Jena US LLC, Upland, CA, USA). The product size of 857 bp and 589 bp was interpreted as *C. jejuni* positive and the product size of 857 bp and 462 bp was interpreted as *C. coli* positive.

### 2.3. LAMP Primers

The LAMP primers used in this study were the primers reported by Yamazaki et al. [19] targeting the *cj0414* gene presumed to encode an oxidoreductase from the sequence of AL111168, as shown in Table 1, which were synthesized by Bio Basic, Markham, ON, Canada.

### 2.4. LAMP Assay and Condition Optimization

The LAMP assay was performed with a LavaLAMP DNA Component Kit (Lucigen, Middleton, WI, USA) using the set of primers discussed earlier. The LAMP assay comprised 2.5 µL of 10× LavaLAMP DNA buffer, 1 µL LavaLAMP DNA enzyme, 25 mM each deoxynucleotide triphosphate (dNTP; SibEnzyme, Novosibirsk, Russia), 100 mM magnesium sulfate, 2 μM F3 and B3, 16 μM FIP and BIP, 8 μM LF and LB, 1× green fluorescent dye (Lucigen, Middleton, WI, USA), and 1 µL DNA template of *C. jejuni* ATCC33291. The final volume was adjusted to 25 µL. No target control (NTC) and positive control (provided in the kit) were used to find the optimal temperature of the LAMP assay. A reaction with 1 µL nuclease-free water instead of a DNA template served as a NTC in each run.

Temperature optimization of the assay was performed in a CFX96 real-time PCR machine (Bio-Rad, Berkeley, CA, USA) and a gradient step for the temperature was set between 68 °C and 74 °C, to monitor reaction fluorescence using the FAM channel to detect amplified product. The real-time condition was set to 60 min (30 s/cycle). The optimal temperature for the LAMP assay was selected based on high differences between *C. jejuni* culture-positive samples and NTC mean C_q_ values.

Real-time LAMP assay optimization with 24 *C. jejuni* positive, 24 NTC, and 2 positive control samples showed that optimal temperature ranged from 69.2 °C to 71.8 °C (Table 2). A temperature of 70.5 °C was chosen by finding the median ranging from 69 °C to 72 °C and the resulted temperature was repeated for confirmation before being used as the optimal temperature in this study.

### 2.5. Receiver Operating Characteristic (ROC) Curve Analysis

*C. jejuni* reference strain and nuclease-free water were used as positive and negative controls, respectively. C_q_ values from 116 *C. jejuni* samples and 113 NTC samples obtained from the real-time LAMP were analyzed for the ROC curve using the SAS University Edition program (SAS Institute Inc., Cary, NC, USA). Statistical modeling was designed based on the C_q_ data from the real-time LAMP to predict cutoff values. The PROC LOGISTIC procedure was performed (Appendix A). The MODEL specified POSITIVE as the dependent variable (left side of the equal sign). By default, SAS modeled the probability of POSITIVE = ‘0’. As a result, an EVENT = ‘1’ option was required, as positive results were defined in the dataset as ‘1’. The CYCLE (C_q_) is the continuous independent variable (on the right side of the equal sign) from which the cutoff score for a positive sample was determined. The OUTROC option in the model produced a second dataset called ROCDATA, which contained sensitivity and specificity data. The ROC statement generated a ROC curve, and the ROCCONTRAST command generated a ROC curve significance test. The area under the curve (AUC) of the ROC curve is a statistical measure of a test’s overall ability to differentiate between two outcomes (Figure 1). The OUTROC option was used to construct a new data set, referred to as ROC2. A new variable, LOGIT, was created using a LOG() function and referred to the log of the odds (the probability of success; _prob_) over the probability of failure (1-_prob_). The CUTOFF variable was formed by rearranging the logistic regression model, logit = intercept + slope(C_q_), to solve for C_q_, as shown in the second part of the analysis (Appendix A). The values of intercept and slope obtained from the PROC LOGISTIC procedure (Table 3) were input into the model for computing each probability cutoff point in the ROCDATA dataset. The fourth and the fifth lines provided sensitivity and specificity of the analysis. The optimal cutoff value was selected based on Youden’s J (YJ) statistic obtained from the model [33]. The YJ is the sum of sensitivity + specificity − 1 and is frequently used as a cutoff value selection criterion. The PROC SORT procedure was used to sort the new ROC2 dataset by YJ in descending order using the DESCENDING option. The last part of the code (PROC PRINT procedure) showed values of cutoff, sensitivity, specificity by the descending sort of YJ. Without regard for biological significance, a value equating to YJ is frequently employed. The C_q_ value was chosen from the maximum YJ without regard for sensitivity or specificity.

### 2.6. Determination of Specificity, Sensitivity, and Limit of Detection of LAMP Assay

DNA of pure cultures of the six bacterial reference strains was prepared as described in the section “Bacterial Strains, Culture Conditions, and DNA Preparation”. Eight samples of each non-*C. jejuni* reference strain were prepared and the total number of non-*C. jejuni* samples was forty-eight. Additionally, a total number of thirty-two *C. jejuni* samples was prepared from the reference strain. The DNA templates were used to assess the sensitivity and specificity of the LAMP assay, which was compared with the standard culture procedure. The cutoff value obtained from the model was used to interpret the sample’s test result from the LAMP assay. C_q_ values less than or equal to the cutoff were interpreted as positive, while values more than the cutoff were interpreted as negative. Sensitivity (SE) and specificity (SP) of the assay were compared with the results from a standard culture procedure using the formula; SE = [TP/(TP + FN)] × 100 and SP = [TN/(TN + FP)] × 100, where the terms true positive (TP) or true negative (TN) were used to express agreement, and false positive (FP) or false negative (FN) were used to indicate disagreement of the two assays.

To perform a limit of detection test, the turbidity of 0.5 McFarland units of *C. jejuni* pure culture was measured using a DEN-1B McFarland densitometer (Biosan, Riga, Latvia), 10^8^ CFU/mL was prepared corresponding to the initial concentration and this initial concentration was 10-fold serially diluted using 1 mL PBS for DNA extraction. Additionally, 100 µL of pure culture dilutions ranging from 10^−2^ to 10^−4^ were used to confirm the amount of *C. jejuni* by a direct plating method using mCCDA plates with the condition as described above. DNA templates of each dilution were used to assess the detection limit of the LAMP assay, which was compared with an mPCR assay. The cutoff value obtained from the model was used to interpret the sample’s test result from the LAMP assay, as described above. Each experiment for the limit of detection tests was conducted in triplicate, and the detection was based on the number of positives, for instance, if all three samples tested positive, the test would be considered positive.

### 2.7. Detection of C. jejuni in Chicken Meat Samples

Chicken meat samples were purchased from GMP-certified chicken slaughterhouses where the samples were collected from *Campylobacter* negative batches and tested negative for *C. jejuni*. The meat samples were preserved at −18 °C and defrosted at 4 °C overnight before being used.

A pure culture of 0.5 McFarland unit of *C. jejuni* reference strain was prepared and diluted to a concentration of 10^6^ CFU/mL to be used for contaminating *C. jejuni*-free chicken meat samples. A 10-fold serial dilution of 10^−3^ to 10^−5^ of 0.5 McFarland unit of *C. jejuni* reference strain was used to quantify the amount of *C. jejuni* by the direct plating method in which dropping 100 μL of each dilution on mCCDA plates, and then those plates were incubated at 41.5 °C for 42 h. Colony count calculations were performed based on the recommended ISO 10272-2:2006 [34].

Two chicken meat samples were prepared as described and 3 replicates was performed for each sample. A prepared bacterial solution of 10^6^ CFU/mL was spiked on cut chicken meat, sized 12 × 12 × 2.5 cm, weighing 250 g, by spiking 1 mL of the solution on each of the front and the back surfaces of the meat and incubating at room temperature for 15 min. A cut of 25 g of chicken meat was mixed with 225 mL supplemented Bolton broth and homogenized with a BagMixer 400 CC stomacher (Interscience, Saint Nom la Brétèche, France) for 2 min. The sample was then incubated at 37 °C for 6 h, and at 41.5 °C for 42 h in a microaerobic atmosphere as described earlier and followed the standard procedure of ISO 10272-1:2006. A 1 mL of the sample at points 1-8 was kept for further detection of *C. jejuni* at the following intervals: 6 h, 12 h, 18 h, 24 h, 30 h, 36 h, 42 h, and 48 h after incubation. A 1 mL Bolton broth sample was directly used for DNA extraction. At point 9, a single loopful of the culture was streaked on a mCCDA agar, then incubated at 41.5 °C for 42 h in a microaerobic atmosphere. Three isolated colonies from a sample were transferred to 1 mL PBS for DNA extraction. DNA extraction was performed using a commercial DNA extraction kit as described in Section 2.1.

### 2.8. Data Analysis

Sensitivity, specificity, and predictive values of the LAMP assay compared with the bacterial culture were analyzed. Additionally, Cohen’s kappa (k) was performed to compare agreement between the LAMP assay and the standard culture using IBM SPSS Statistics version 28.0.1.0 (Armonk, NY, USA).

## 3. Results

### 3.1. ROC Curve Analysis

The output of partial logistic regression is shown in Table 3. The intercept and slope values were 2.3689 and −0.0682, respectively, calculated from the logistic regression analysis, which was utilized to generate the model’s cutoff value. The ROC curve from the analysis is shown in Figure 1, with an AUC of 0.936. The AUC was subjected to significance testing in Table 4. The model indicated that the AUC was significant (95% Wald CI: 0.903–0.970), which was confirmed by Chi-square (*p* < 0.0001).

The logistic regression analysis’s intercept and slope were used to indicate each probability cutoff value in the rocdata dataset. The YJ statistic was employed in the model to sort possible cutoff values with a high degree of sensitivity and specificity. A C_q_ value of 18.07 was used as the cutoff value in the LAMP assay to differentiate between positive and negative test results in this study based on the model’s highest YJ, sensitivity, and specificity (Table 5).

### 3.2. Specificity, Sensitivity, and Limit of Detection of LAMP Assay

The specificity test of the LAMP assay using the C_q_ cutoff value obtained from the model revealed that the cutoff value correctly identified 96.9% (31/32 samples) of positive-*C. jejuni* samples and 100% negative results were shown for *S.* Paratyphi, *P. aeruginosa*, *S. aureus*, and NTC (0/8 samples). However, the given cutoff value could not classify one of the eight samples of *C. coli* and *E. coli*, resulting in a 12.5% false-positive (Table 6). Compared to the mPCR assay, the LAMP assay correctly showed results of positive *C. jejuni* samples (100%) and negative results with *C. coli* (0%). However, the primer set used in mPCR cannot differentiate *S.* Paratyphi, *P. aeruginosa*, *S. aureus*, and *E. coli* (100% false-positive). Compared with a standard culture method, our LAMP assay showed 96.9% sensitivity and 95.8% specificity, and the positive and negative predictive values were 93.9% and 97.9%, respectively (Appendix A). The k statistics indicated the agreement between the LAMP assay and the culture method was 0.92 (*p* < 0.001).

Direct plating of 0.5 McFarland for the initial *C. jejuni* counts showed that the starting number of *C. jejuni* was 4.8 × 10^8^ CFU/mL. Comparing the sensitivity of the LAMP with the mPCR assay (Table 7), we could identify a low number of *C. jejuni* at 10^3^ CFU/mL in the LAMP assay, whereas mPCR detected less than 10 CFU/mL.

### 3.3. Detection of C. jejuni in Chicken Meat Samples

DNA templates, from chicken meat containing 10^6^ CFU, collected at nine points during the incubation process, demonstrated that the LAMP assay test produced more fluctuating results than the mPCR assay for the detection of *C. jejuni*. In the LAMP assay, samples obtained at various intervals throughout the incubation period yielded either negative results or only one positive result out of three replications. In the mPCR assay, DNA templates isolated from colonies on mCCDA produced positive results on gel electrophoresis by analyzing all replicating samples (Table 8).

## 4. Discussion

### 4.1. LAMP Assay Optimization

The LAMP assay used in the study was modified from Yamazaki et al. [19] by optimizing the reaction temperature to be compatible with the commercial amplification kit. The gradient temperature setting on the real-time instrument was adjusted to the amplification kit manufacturer’s recommended range of 68 to 74 °C. The selection of the optimal temperature was based on the temperature that provided high differences of C_q_ between *C. jejuni* positive samples and the NTC. In this experiment, based on the real-time gradient temperature setting, it was found that the temperature range between 69.2 and 71.8 °C was optimal. Thus, the median of the ranged temperature (70.5 °C) was chosen.

### 4.2. ROC Curve Analysis

This study was the first study that used ROC curve analysis to find the optimal cutoff value of C_q_ to use in the LAMP assay. The regression output of the ROC model indicated that the quantification cycle was significant (*p* < 0.001). According to the ROC model, the region in the top left (Figure 1) offered the most beneficial discrimination in terms of a cutoff score. The model’s ROC curve was aligned above the diagonal line, which ran from (0,0) to (1,1), representing the random chance line or line of equality [35]. Values plotted above and to the left of the line of equality represented correctly predictive results and indicated a more sensitive test (greater true positive rate). This implied that the C_q_ could not differentiate between positive and negative samples by chance. In this study, the AUC of 0.936 indicated that of all potential positive/negative sample pairings generated by the tests, the model with C_q_ projected a greater anticipated likelihood of being positive for *C. jejuni*-positive samples than for negative samples (93.6% of the time). The model showed the significance of AUC (95% CI: 0.903–0.970) in terms of the classification of positive samples, making it possible to determine an optimal C_q_ cutoff value for *C. jejuni* positives. A Chi-square test also confirmed the significance of the model (*p* < 0.0001). To summarize, C_q_ accurately identified 93.6% of the time randomly generated pairs of positive/negative *C. jejuni* samples.

There is one aspect to consider when the cutoff was chosen based on the model’s highest YJ calculation. A disadvantage of the YJ is that it is incapable of distinguishing between variations in sensitivity and specificity [35]. Although similar YJ was observed from both tests, it is not inferable to the same sensitivity and specificity, as YJ is the sum of sensitivity and specificity minus one. This can result in the two tests performing differently; for example, the first test has a sensitivity and specificity of 0.7 and 0.9, respectively, whereas the second test has a sensitivity and specificity of 0.8 and 0.8, respectively. All three model parameters (YJ, sensitivity, and specificity) are considered for the selection of a cutoff value. In this study, the maximum YJ value served as the cutoff for the prediction model. According to the model, the ideal cutoff value with the highest YJ (0.808) was 18.07, which provided the best combination of sensitivity and specificity. The cutoff value (C_q_) of 18.07 indicated that the test required about 9 min to differentiate between positive and negative samples. When interpreting the results using end-point techniques, such as UV light fluorescence, agarose gel, or turbidity test, the cutoff value of C_q_ relates to the amount of time spent on the test or the duration of incubation time for reactions. Incubation time should not go beyond this point to avoid a false-positive interpretation that can occur from non-specific amplification due to the high concentration of primers commonly used in the assay. Compared with the study of *cdtC*–*gyrA* LAMP assay, the mean detection time of the assay in the detection of *C. jejuni* was 12 min [36].

### 4.3. Non-Specific Amplification of LAMP Assay

When the number of samples was increased, some negative samples with low C_q_ values on real-time LAMP were observed, resulting in a low cutoff from the ROC modeling. The low number of C_q_ obtained from the model indicated a low testing time used in the assay. Compared to other LAMP assays, the general procedure of reaction incubation at a specific temperature was set to 60 min [19,37]. Due to the higher concentration of primers used in LAMP than in traditional PCR methods, non-specific amplification of false positives induced by primer dimers is possible [38]. The LAMP assay required a 10× concentration of the primers, which accounts for 10% of reaction volume, whereas the mPCR assay required a 1× concentration of primers. According to Wang’s study, adding dimethyl sulfoxide (DMSO) at low concentrations to the reaction mix enhances amplification by shortening the time required for detection and inhibits non-specific amplification by suppressing DNA polymerase activity [39]. Wang’s study incorporated a modified method known as Touchdown LAMP by preheating the reaction at 95 °C for 5 min and then gradually decreasing the reaction temperature from high (+6, +4, +2 °C, respectively) to the designated temperature. This method demonstrated increased sensitivity and specificity compared to the conventional LAMP assay [39]. A preliminary evaluation of the effectiveness of Touchdown LAMP was performed in this work to mitigate the effect of non-specific amplification from primer dimers. Nonetheless, no significant result was observed (data not shown). Additionally, the LAMP assay was created for isothermal operation, which necessitated the use of basic equipment, such as a water bath or heat block. Hence, a modification Touchdown LAMP technique does not meet the purpose of the study.

In this study, non-specific amplification of LAMP reactions was unintentionally observed while running the reaction that was left at room temperature without the addition of DNA templates, showing a positive banding pattern on agarose gel (data not shown). Additionally, adding primers immediately before the addition and incubation of the target DNA, as well as directly transferring reactions from ice to a heat block or thermal cycler at the correct reaction temperature, are suggested to avoid the interaction of substances before the specified conditions. Melt curve analysis in a real-time assay was recommended to detect amplified products to determine non-specific amplification [38]. In addition, a restriction fragment analysis of LAMP results was performed in the study by Babu when there is insufficient amplifiable DNA to produce a visible color change in the reaction mixture. It provided a reliable secondary confirmation of the LAMP reaction and clearly distinguished false positives from true positives [40]. After running on an agarose gel, non-specific or background amplification appeared as a smear of DNA fragments with no apparent or identifiable bands. It is possible that DNA contamination occurred as a result of shared reaction preparation and sample loading areas. To avoid contaminating LAMP reactions with target DNA or target amplicons, it is recommended to identify and use a reaction setup area that has never been exposed to target DNA or amplified products [41]. Additionally, it is advisable to use a second location that has never been exposed to amplified material to add template DNA to reactions and to designate a third area for analyzing LAMP reaction products to avoid reaction contamination [42]. Moreover, a cold reaction setup is required for the LAMP assay. The commercial DNA enzyme has been indicated that it retains activity beyond 4 °C, which might result in non-specific background amplification in the subsequent reaction. Furthermore, a closed tube LAMP assay was developed in the study by Karthik et al. with the use of agar dye capsules to avoid product cross-contamination, which might result in false-positive findings [43].

### 4.4. Campylobacter Species Identification with mPCR Assay

Bacterial culture with a mPCR assay confirmation was employed in this work as a conventional approach for identifying *C. jejuni* and *C. coli*. Discrimination between the closely related species *C. jejuni* and *C. coli* is phenotypically based only on the Hippurate hydrolysis test [44]. Compared to biochemical testing, a mPCR assay using primers consistently differentiated the two major *Campylobacter* species: *C. jejuni* and *C. coli* (100% specificity). Nevertheless, discriminating of other foodborne pathogens, a *mapA* gene targeting *C. jejuni* yielded false-positive results for several non-target strains (Table 6), reducing its specificity (33%). The low performance of a *mapA* gene-based PCR was also reported in other findings [45]. However, there was no prior report on the *mapA* gene’s specificity when tested against the non-*C. jejuni* foodborne bacterial strains that were used in this study. A further study on the effectiveness of a *mapA* gene-based PCR and the parameters associated with low specificity is highly recommended. In this study, primers specific for the *mapA* gene provided accurate findings for the identification of two *Campylobacter* species compared to results validated by microbiological methods. Among the examined foodborne bacterial strains, *Campylobacter* is the only one that requires particular growth conditions due to its requirement for a microaerobic atmosphere. Consequently, before DNA extraction for the detection of *C. jejuni*, a culture procedure with specific enrichment and selective media, as recommended by ISO 12072-1:2006, should be performed to identify *Campylobacter* bacteria, as illustrated in Figure 1, to avoid false-positive gel electrophoresis results. The study by Kabir et al. suggested using a *hipO* and *ask* gene-based PCR assay for high sensitivity and specificity in detecting *C. jejuni* and *C. coli*, respectively. However, significant differences in annealing temperature and time required for amplification, as well as the number of cycles should be considered for a mPCR assay [45]. Additionally, a combination of *cdt* gene-based mPCRs, which is performed using similar PCR conditions, was also recommended due to its efficacy on species identification for both *C. jejuni* and *C. coli* [45,46].

### 4.5. Specificity and Sensitivity of LAMP Assay

A primer set used in the LAMP assay was obtained from the sequence of the *cj0414* gene, which encodes putative glucose–methanol–choline oxidoreductase subunits [19,47]. The primer set was chosen based on its high specificity for the identification of *C. jejuni*, as no previously reported primer set based on the *ceuE*, *hipO*, *mapA*, or 23S rRNA gene has demonstrated 100% accuracy [48]. In this study, using the cutoff determination of the samples resulted in false positives, in some samples of *C. coli* and *E. coli* strains. In contrast, no non-*Campylobacter* bacteria tested positive with the assay in our referent study [19]. Considering only *C. jejuni* detection, we modified the LAMP assay by optimizing the reaction temperature and using the cutoff value for interpretation, to show higher sensitivity and specificity (96.9% and 95.8%, respectively) in this study, which was compared to the study by Yamazaki et al. (2008) showing 79.3% for sensitivity and 91.8% for specificity [19]. Cohen’s kappa (k) statistics showed an excellent agreement between the LAMP assay and the standard culture procedure (k = 0.92, *p* < 0.001), indicating the reliability of the assay [49]. 

From our review, a designed set of primers targeting the *rplD* gene provided 100% specificity in the detection of *Campylobacter* (both *C. jejuni* and *C. coli*), and the *cdtC* gene- and *gyrA* gene-based duplex LAMP provided simultaneous detection of *C. jejuni* and *C. coli*, using melting curves for the differentiation with 100% specificity when performed using 62 *C. jejuni*, 27 *C. coli*, and 85 non-target species in the study by Kreitlow, 2021 [36]. To date, there are studies that have taken advantage of the LAMP assay’s rapidity and simplicity to develop primers that allow the simultaneous detection of multiple bacterial strains, such as the development of degenerate primers targeting the 16S rRNA sequence for the all-inclusive detection of human pathogenic *Campylobacter* species (*C. jejuni*, *C. coli*, and *C. lari*) or the development of duplex LAMP assays for the identification of two bacterial strains [36,40,50].

### 4.6. Limit of Detection of LAMP Assay

The LAMP assay in this study can detect *C. jejuni* at a lower level of 10^3^ CFU/mL, which is in line with Yamazaki’s (2008) investigation using the same set of primers [19]. The detection limit of the LAMP assay was 100 times higher than the equivalent PCR assay. The studies of Kreitlow and Sridapan also reported the detection limit of 10^3^ CFU/mL in enriched meat samples using the LAMP assays targeting the *cdtC*–*gyrA* genes (*cdtC* primer set for the detection of *C. jejuni* and *gyrA* primer set for the detection of *C. coli*) and 16S rRNA, respectively [36,50]. In another study by Yamazaki, it was reported that Preston enrichment media was used for the growth of *C. jejuni* contamination in chicken meat samples in combination with a three-step centrifugation procedure, reduced culture time, and increased detection sensitivity [51]. In that study, the three-step centrifugation techniques were used to remove larger debris from chicken samples and components of the Preston enrichment broth that included DNA amplification inhibitors, as well as to concentrate the small number of *Campylobacter* cells. The results of Yamazaki’s study indicated that *C. jejuni* had a sensitivity of 8 CFU per LAMP reaction tube, corresponding to 4.8 × 10^2^ CFU/mL [51]. In the study by Quyen, a Loopamp *Campylobacter* detection kit was used for the detection of *Campylobacter* in feces [52]. Without enrichment, the assay was able to detect the presence of *Campylobacter* as low as 50 CFU/mL in fecal samples, which required a testing time of more than 30 min. However, the detection kit has its limitations on specifically identified *C. jejuni* or *C. coli* [52].

### 4.7. Detection of C. jejuni in Chicken Meat Samples

In this study, using the LAMP assay, the cultivation step was omitted for chicken meat samples streaked on mCCDA and CBA, which resulted in negative findings. The viable but non-culturable (VBNC) condition is thought to present one of *Campylobacter*’s survival mechanisms against harsh environmental stresses, such as nutrient starvation, extreme temperatures, osmotic stress, oxygen availability, and chemicals [53]. The VBNC state is characterized as a dormant condition in which bacterial cells retain active metabolism but are unable to be cultivated on standard bacteriological media [54]. Specialized nutrients and conditions are required for the growth of *C. jejuni* due to their unique properties, so the process of enrichment was not restricted to minimize the time of testing. However, the study by Petersen demonstrated the induction of VNBC *C. jejuni* using the osmotic stress with a 7% (*w*/*v*) NaCl solution. An assay combining a LAMP and propidium monoazide (PMA-qLAMP) was developed for the elimination of DNA amplification signals from dead bacterial cells. A standard curve of a known concentration was generated for the quantification of VBNC *C. jejuni* in the spiked chicken breast meat samples, with a range of detection from 3.78 × 10^2^ to 3.78 × 10^5^ CFU/g [54]. The PMA-qLAMP assay may aid in the surveillance of potential *C. jejuni* in the food production chain coupled with a culture procedure.

Additionally, prolonged stomaching of meat samples was noted to enhance the release of inhibitory factors from chicken meat samples, such as organic and phenolic compounds, glycogen, fats, and calcium ions. Light hand massaging of no more than 30 s is preferable since it results in the low release of inhibitory factors [41,55]. While the LAMP assay aids in reducing testing time, the conventional culture procedure is a necessary step that cannot be omitted due to the assay’s limited sensitivity. The use of Preston enrichment broth, as reported in Yamazaki’s (2009) work, as well as a three-step centrifugation process for DNA extraction, were proposed for further evaluation of the results obtained from the LAMP assay. In addition, the enrichment of samples taken from chicken meat with Bolton broth and the application of a three-step centrifugation procedure for DNA extraction were also mentioned to yield high sensitivity. The preparation of DNA using 1 mL Bolton enrichment culture, followed by three-step centrifugation which yielded 100% specificity and 100% sensitivity was compared with that of the template DNA following the instructions of the manufacturer which yielded 95.2% specificity and 76.2% sensitivity [51]. Detection of this bacteria required more than 4 days for isolation and identification as followed by the standard culture protocol provided in this study (Figure 1). Rodgers’ study found that Preston broth and Bolton broth were significantly less effective in detecting *Campylobacter* (*C. jejuni* and *C. coli*) compared with other enrichment media for sensitivity and accuracy estimation of *C. jejuni* incidence in broiler flocks at slaughter from cecal samples [56]. Enrichment in Exeter broth with a resuscitation step was found as the most effective detection method with 95.7% sensitivity to *C. jejuni*, whereas using adapted Exeter broth (deficient in polymyxin B) with a resuscitation step was 100% sensitive for the detection of the genus *Campylobacter*.

### 4.8. Limitations

Sample preparation with an enrichment step is a limitation of this LAMP assay on rapid detection strategies that are aimed to be used for the poultry production chain, as also seen in the study by Kreitlow [36]. The total time from sample preparation to the final result was still too long compared with the kind of sample used, such as feces. There were some studies that were capable of directly extracting the DNA from a prepared sample [52,57]. The initial concentration of the bacteria was suggested to be an important factor that affected the time spent on the sample preparation step. When samples contained a high concentration of *C. jejuni*, such as feces, enrichment procedures to isolate these bacteria from feces are typically unnecessary [41]. Additionally, characteristics of sample types should be further studied for a better understanding in order to find a proper method of sample preparation that can reduce the time of that step before proceeding to the testing step. According to the experiments in this study, the cost of testing using the LAMP assay was approximately USD 9.28 per reaction, including DNA extraction, and the total cost of the whole process (including enrichment) was USD 36.21 per sample. The cost of each test is a primary concern for the application of the test on-site. Adjusting the reaction volume and optimization of primer concentration were performed and discussed in the study by Quyen. The results demonstrated that there was no effect of the reaction volume on the LAMP efficiency or assay principle [52].

This study was a preliminary study for the application of rapid assay at field operations by optimizing the referenced LAMP assay to be compatible with the commercial amplification kit available. To control the quality of DNA templates, a DNA extraction kit was used, which required at least 1.5 h for the extraction process. DNA extraction methods, such as boiling, treating with NaOH, and other rapid DNA extraction methods are suggested for decreasing the time of DNA template preparation to less than 30 min [51,58]. A limitation of the study was the number of bacterial strains used. To analyze the efficiency of the LAMP assay, it is recommended to increase the number of non-*Campylobacter* strains as well as the number of *Campylobacter* field strains for a more precise evaluation of sensitivity and specificity of the assay.

## 5. Conclusions

This study provided a practical idea of LAMP assay application by optimizing reaction temperature to be compatible with the commercial LAMP reaction kit. Although the cutoff value used in this study resulted in a short time for *C. jejuni* detection, the LAMP assay with the specified primer set is recommended for a screening test due to the limitation of sensitivity and specificity demonstrated. Confirmation of the positive samples with a standard method or other techniques is required. The advantages of this assay are requiring less time and fewer instruments compared with the standard culture method and a molecular technique, such as PCR. Thus, our proposed assay is applicable at any processing steps in the food production chain, as it takes less than 3 h to give the final result, including the DNA extraction process. Further investigations might concentrate on developing a new primer set for more accuracy in the detection of *C. jejuni* and reduction in the time used in an enrichment step.

## Data Availability

The data presented in this study are available on request from the corresponding author. The data are not publicly available due to institutional privacy policy.

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
