# Peer review of "A Cutoff Determination of Real-Time Loop-Mediated Isothermal Amplification (LAMP) for End-Point Detection of *Campylobacter jejuni* in Chicken Meat"

_vetsci, 2022, doi:10.3390/vetsci9030122_

Round 1

Reviewer 1 Report

Campylobacter is a major cause of foodborne illness and methods to rapidly detect the bacterium in chicken meat are continually being sought. This study describes the determination of a cutoff value using receiver operating characteristic (ROC) curve modelling for a LAMP assay that could be used to detect Campylobacter in chicken meat. While the data generated in this study may be of value in applying the assay in general terms, there are several major issues with the manuscript which must be addressed.

A major concern is that that although the optimised LAMP assay resulted in robust detection of Campylobacter from pure culture, experimental approaches involving chicken meat were limited and the assay performed poorly for these samples. The method therefore appears to provide no significant improvements in the detection of Campylobacter in food. Either substantial additions to the work are required, demonstrating applicability in analysing food samples or the manuscript should be re-written to focus soley on assessing the technical aspects of the method.

The results section is very confused and requires considerable re-writing with a clearer explanation of what information is being presented throughout, possibly using subheadings for each section.

Another area of concern is the mPCR results (Table 6), where C. jejuni PCR positives were found for several non-campylobacter species. This would cast serious doubt on the application of this mPCR test for DNA extracted from whole samples (faecal/food), rather than colonies from selective Campylobacter plates. This work should therefore be repeated following the authors own recommendations for clean PCR (if possible) and the PCR products should be sequenced to ensure that these false positives were not a result of sample contamination.

The observation that the LAMP assay gave 12.5% false positives (detection of E. coli and C. coli) indicates that the specificity of the test may be much lower when using a biological sample rather than a no-target control (used for the ROC analysis). This is a significant proportion of false positives and should be highlighted in the abstract.

The authors include sufficient background information in the introduction and appropriate references are included in the discussion are but many of the sections are confusing and there are many sentences that require re-writing.

The materials and methods section would benefit from being re-ordered with the ROC curve analysis following section 2.4 (LAMP Assay and Condition Optimisation). The ROC curve analysis requires a fuller explanation. Clearer details on the samples used for this analysis (e.g. which Campylobacter strains were used for this part of the study) are also required as well as more information on the no-target control samples.

For Table 5, the data should be sorted by Cutoff score so the results can be assessed more easily.

More information is required on the preparation of chicken meat, such has how the meat was homogenised. Was the meat tested for the presence of Camplylobacter prior to spiking?

Author Response

Dear reviewer,

We would like to thank the reviewer and we are very grateful that you had dedicated your time for assessing our study.

We have responded to all of your comments. All of the concerns raised by the reviewers were addressed in the revised manuscript.

Regards,

Reviewer 2 Report

Reviewer comments and suggestions

The purpose of this study was to develop loop-mediated isothermal amplification (LAMP) assay with an amplification kit to use as a rapid test for C. jejuni detection to determine an optimal cutoff value of quantification cycle (Cq), obtained from the real-time LAMP, using receiver operating characteristic (ROC) curve modeling. A value acquired from the study is used as a testing time for the screening of C. jejuni at food processing facilities using end-point LAMP assay to achieve precise test results. Efficacies in terms of sensitivity and specificity were also evaluated in this LAMP assay in comparison to a standard culture method. The authors had suggested that this assay could be an alternative screening test for C. jejuni contamination in food samples.

The paper has nicely complied. However, in many places, the authors need to add the relevant references and there were a few inconsistencies in the manuscript. Based on my view, below are the comments that need to be incorporated in the revised version of the manuscript. 

  1. The present studies look similar to reference 19 and 44. Please point out the novelty of your study.
  2. LAMP primers (table 1) used in this study is the same as reference 19 as you have mentioned. Why a new set of primers or targets are not designed? 
  3. Line 140, Paratyphi should be small and italic. 
  4. The specificity of the developed LAMP assay should be checked with more species specifically including C. lari and C. upsaliensis.
  5. Line- 400-401, reference 19 please discuss more about the study.
  6. An important paper published in 2019, A Sensitive, Specific and Simple Loop-Mediated Isothermal Amplification Method for Rapid Detection of Campylobacter spp. in Broiler Production has not been discussed by you in the discussion part. Please discuss the study in the discussion part.
  7. Discussed the result and compare your result with the paper published in 2021 (Combined Loop-Mediated Isothermal Amplification Assays for Rapid Detection and One-Step Differentiation of Campylobacter jejuni and Campylobacter coli in Meat Products )for detection of Campylobacter jejuni and Campylobacter coli in Meat Products.
  8. Discuss this paper (Sensitive and rapid detection of Campylobacter jejuni and Campylobacter coli using loop-mediated isothermal amplification by Wataru Yamazaki, 2013) results in discussion part and compare the outcomes. 
  9. Discuss the following paper in discussion part- https://pubmed.ncbi.nlm.nih.gov/32707152/, https://www.sciencedirect.com/science/article/abs/pii/S0168160521002221 and https://journals.plos.org/plosone/article?id=10.1371/journal.pone.0254029. 
  10. In all references, journal-style should be modified according to the MDPI guidelines.

Author Response

(The authors gave the same response as above.)

Reviewer 3 Report

A Cut-off Determination of Real-Time Loop-Mediated Isothermal Amplification (LAMP) for End-Point Detection of  Campylobacter jejuni in Chicken Meat

Dear author and editor:

The author in this study suggested an optimal method for LAMP reaction in detection procedure for Campylobacter jejuni.

the article could be published after a minor revision:

I have some comments on it:

  • Which is more specific and more sensitive test in Campylobacter jejuni mPCR or LAMP? Please clarify this point in the text.
  • Which kind of samples that the test could be applied for?
  • In your suggestion the author said that this test could be applied to any step in food production process? Please verify more the nature of samples.
  • What is the advantages and disadvantages of using LAMP compared with multiplex PCR?

Thank you very much, best regards

Author Response

(The authors gave the same response as above.)
